# MoveFM-R: Advancing Mobility Foundation Models via Language-driven Semantic Reasoning

## Abstract

Mobility Foundation Models (MFMs) have advanced the modeling of human movement patterns, yet they face a ceiling due to limitations in data scale and semantic understanding. While Large Language Models (LLMs) offer powerful semantic reasoning, they lack the innate understanding of spatio-temporal statistics required for generating physically plausible mobility trajectories. To address these gaps, we propose MoveFM-R, a novel framework that unlocks the full potential of mobility foundation models by leveraging language-driven semantic reasoning capabilities. It tackles two key challenges: the vocabulary mismatch between continuous geographic coordinates and discrete language tokens, and the representation gap between the latent vectors of MFMs and the semantic world of LLMs. MoveFM-R is built on three core innovations: a semantically enhanced location encoding to bridge the geography-language gap, a progressive curriculum to align the LLM's reasoning with mobility patterns, and an interactive self-reflection mechanism for conditional trajectory generation. Extensive experiments demonstrate that MoveFM-R significantly outperforms existing MFM-based and LLM-based baselines. It also shows robust generalization in zero-shot settings and excels at generating realistic trajectories from natural language instructions. By synthesizing the statistical power of MFMs with the deep semantic understanding of LLMs, MoveFM-R pioneers a new paradigm that enables a more comprehensive, interpretable, and powerful modeling of human mobility. The implementation of MoveFM-R is available online at `https://anonymous.4open.science/r/MoveFM-R-CDE7/`.

## 1 Introduction

The proliferation of large-scale mobility data from sources like GPS and location-based services has revolutionized the modeling of human mobility (Luca et al., 2021; Feng et al., 2018; Yuan et al., 2025; Chen et al., 2024), which is a foundational element of human behavior and the engine of urban functionality (Gonzalez et al., 2008; Song et al., 2010). The field has witnessed a remarkable architectural evolution, progressing from early statistical approaches (Kitamura et al., 1996; Arentze et al., 2000; Bowman & Ben-Akiva, 2001) to sophisticated deep learning frameworks (Feng et al., 2018; Yang et al., 2022; Yuan et al., 2023; Li et al., 2024; Chu et al., 2023; Zhu et al., 2024a).

Inspired by the pursuit of Artificial General Intelligence, the foundation model paradigm has recently been introduced to the domain of human mobility (Zhou et al., 2024). A new line of research has focused on building mobility foundation models (MFM) from scratch (Zhu et al., 2024b; Han et al., 2025; Liu et al., 2024b; Long et al., 2025), which have demonstrated remarkable generalization capabilities across a variety of tasks and contexts. Despite their impressive performance, a fundamental ceiling remains, stemming from two core issues. On the one hand, the scale of available mobility data, though large, is constrained by privacy concerns and collection costs (Kim et al., 2020). It is dwarfed by the almost unimaginable scale of web data that fuels LLMs, making it difficult to replicate their emergent intelligence from scratch. On the other hand, these models effectively process geographic coordinates but cannot infer the rich semantic context and human intent that drive these mobility patterns.

However, we argue that simply replacing MFMs with LLMs is not the answer, as LLMs are not "native speakers" of the continuous, physically-constrained movement; they lack the deep, built-in understanding of spatio-temporal statistics and distributions that specialized MFMs excel at. Current LLM-based models (Shao et al., 2024a; Wang et al., 2024) struggle to ground their reasoning in physical reality; they can produce sequences of plausible location types, but these sequences are often geographically incoherent or physically infeasible (Koda et al., 2025). The optimal path forward is therefore synthesis, not replacement. Building on this premise, our work leverages the unique semantic reasoning of LLMs to fully unlock the potential of MFMs, thereby addressing their core limitation in semantic understanding. Furthermore, this paradigm enhances usability, as natural language provides a more intuitive and expressive interface for guiding the generation process (Reynolds & McDonell, 2021). For example, the instruction can be like "please generate xxx".

This proposed synthesis, while promising, faces two fundamental challenges. The first is a fundamental vocabulary mismatch. Natural language processing benefits from a finite, shared vocabulary, whereas mobility unfolds across a near-infinite and continuous set of locations. Simply discretizing coordinates leads to an explosive vocabulary size and loss of precision (Chen et al., 2025). Second, a significant representation gap exists between the two modalities. An MFM's understanding of mobility is expressed through latent vectors that capture the statistical and geometric patterns of movement (Hashemi & Zufle, 2025). These representations, however, are not directly interpretable by an LLM, which reasons about the world through the lens of human language and semantics (Singh et al., 2024).

To address these challenges, we propose MoveFM-R, which unifies the mobility understanding of MFMs with the semantic understanding and reasoning capabilities of LLMs. Its design philosophy is to bridge the mismatch between continuous trajectories and discrete language, making it easier for LLMs to understand the spatiotemporal features of MFM trajectories. First, MoveFM-R introduces semantically enhanced location encoding to discretize continuous coordinates into a set of compact, interpretable tokens, alleviating the vocabulary explosion problem and embedding geographic semantics in a form that LLMs can understand. Second, the description-to-summarization process gradually integrates LLM with movement representation generated by MFM, transitioning from fine-grained natural language trajectory descriptions to higher-level summaries, thereby enhancing its understanding of mobility behaviors. Finally, a self-reflective reinforcement learning strategy iteratively improves the generated trajectories under spatiotemporal constraints, ensuring their plausibility and adaptability in diverse scenarios. These designs collectively address core challenges, enabling MoveFM-R to seamlessly integrate statistical modeling of human movement with semantic reasoning. Our key contributions are summarized as follows:

- We pioneer a novel paradigm to synthesize the statistical modeling capabilities of MFMs with the powerful semantic reasoning of LLMs, enabling a more comprehensive mobility modeling.

- We propose MoveFM-R, a framework built on three core innovations: a semantic location encoding to bridge the geography-language gap, a progressive curriculum to align the LLM with mobility patterns, and an interactive self-reflection mechanism for conditional generation.

- We demonstrate state-of-the-art performance on mobility prediction and generation through extensive experiments, showing significant improvements over MFM and LLM baselines, robust zero-shot generalization, and high-fidelity generation from natural language instructions.

## 2 RELATED WORKS

### 2.1 BUILDING MOBILITY FOUNDATIONAL MODELS FROM SCRATCH

The availability of large-scale trajectory data has facilitated the development of foundational models for human mobility. Early work, such as the Pretrained Mobility Transformer (PMT) (Wu et al., 2024), demonstrated that large-scale pre-training can capture transferable, region-independent movement patterns. Subsequent research has expanded this paradigm, including enhancing cross-city transfer capabilities (Kang, 2025), exploring generative frameworks such as diffusion models (Chu et al., 2023), and leveraging mixture-of-experts (MoE) architectures for improved scalability (Zhu et al., 2024b; Liu et al., 2024b; Shi et al., 2024; Han et al., 2025). Despite their success in modeling statistical patterns, these models operate on coordinate and sequence-based language and

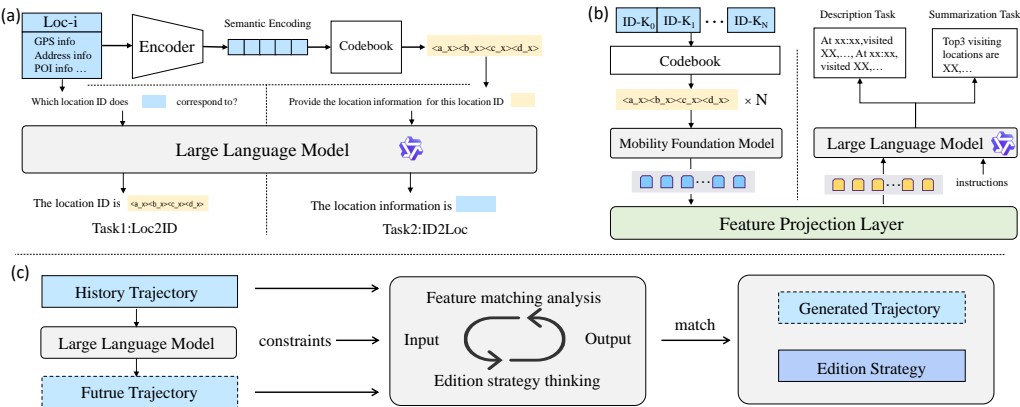

Figure 1: The framework of MoveFM-R. (a) Semantic enhanced location encoding, (b) Mobility understanding from description to summarization, (c) Interactive mobility generation

lack inherent mechanisms for understanding high-level semantics and human intent. This fundamentally limits their reasoning capabilities and motivates the integration of large language models.

## 2.2 LLM-BASED MOBILITY MODELING

Recently, researchers have explored the application of large language models (LLMs) to the mobility domain. Through specialized codebooks or sequence reprogramming (Gong et al., 2024; Chen et al., 2025; Chib & Singh, 2024), continuous trajectories are aligned with the discrete input space of the LLM. On the generative side, researchers have encouraged LLMs to simulate human decision-making processes (Shao et al., 2024a) or act as urban agents to generate trajectories (Wang et al., 2024; Ju et al., 2025). Another approach is to enrich the original trajectories with semantic attributes, such as points of interest (POIs) or activities, to improve model performance (Luo et al., 2024; Liu et al., 2024a; Lan et al., 2024). While these studies successfully incorporate semantic knowledge into mobility modeling, they must reduce continuous trajectories to discrete sequences of symbols in order to make spatiotemporal data digestible to LLMs. This process typically sacrifices geometric accuracy and can produce trajectories that are semantically plausible but geographically incoherent or physically unfeasible. Our research project, MoveFM-R, directly addresses this fundamental challenge by tightly integrating the semantic reasoning capabilities of LLMs with the statistical fidelity of a dedicated mobility encoder, aiming to achieve the best of both worlds.

## 3 METHODOLOGY

To bridge the gap between the statistical power of MFMs and the semantic reasoning of LLMs, we propose MoveFM-R. As illustrated in Figure 1, MoveFM-R progressively integrates mobility patterns with the LLM via three core stages: **(a) Semantically Enhanced Location Encoding**, which translates complex geographic location information into a discrete, semantically rich vocabulary for the LLM; **(b) Mobility Understanding from Description to Summarization**, which enables the LLM to comprehend spatiotemporal patterns through a curriculum progression; and **(c) Interactive Mobility Generation**, which empowers the LLM to iteratively refine and generate realistic trajectories under specified instruction constraints.

## 3.1 SEMANTIC ENHANCED LOCATION ENCODING

LLMs inherently lack an understanding of raw geographic coordinates. To address this, we transform discrete locations into a semantically rich vocabulary by discretizing a high-dimensional geographic semantic space (rather than the raw coordinate space). This core design captures the functional and contextual essence of locations. Furthermore, because the language used to describe geographic concepts is largely universal, this semantics-first approach naturally creates a unified codebook that generalizes across different cities. This process involves two stages: (i) Universal Codebook Construction; and (ii) Codebook Alignment with LLM.

### 3.1.1 UNIVERSAL CODEBOOK CONSTRUCTION

Our approach begins by establishing a common vocabulary that adheres to a semantics-first principle. To achieve this, we first compile a comprehensive semantic profile for each location within a large-scale, multi-city dataset. Rather than relying solely on coordinates, each profile aggregates diverse textual attributes, including street addresses and 34 types of nearby Points of Interest (POIs), which are sourced from OpenStreetMap (OSM). These rich, descriptive profiles are then encoded into high-dimensional semantic vectors using a pre-trained text encoder (Zhang et al., 2025). For detailed information about textual attributes profile, please refer to the appendix B.

The next step is to discretize these vectors into a compact and structured vocabulary. To achieve this, we employ a Residual Quantized Variational Autoencoder (RQ-VAE) (Lee et al., 2022), a powerful technique for high-fidelity vector quantization. The RQ-VAE performs hierarchical quantization, decomposing each semantic vector into a sequence of discrete codewords in a cascaded manner.

Formally, given an input semantic vector $E = r_0 \in \mathbb{R}^d$, the process iteratively quantizes a residual vector at each of the $N$ layers. At the $n$-th layer, a codeword vector $v_{c_n}^n$ is selected from the layer's codebook $\mathcal{C}^n$ as the nearest neighbor to the current input residual $r_n$:

$$v_{c_n}^n = \arg \min_{v \in \mathcal{C}^n} \|r_n - v\|_2^2, \tag{1}$$

where the residual for the next layer is calculated as $r_{n+1} = r_n - v_{c_n}^n$. This decomposes the original vector $E$ into a sequence of indices $\{c_1, c_2, \ldots, c_N\}$, which serves as its discrete representation.

To optimize the codebook, we employ two complementary losses. The residual quantization loss, $\mathcal{L}_{\text{RQ}}$, encourages each codebook to accurately represent the input residuals:

$$\mathcal{L}_{\text{RQ}} = \sum_{n=1}^{N} \left( \|\text{sg}[r_n] - v_{c_n}^n\|_2^2 + \alpha \|r_n - \text{sg}[v_{c_n}^n]\|_2^2 \right), \tag{2}$$

where $\text{sg}[\cdot]$ is the stop-gradient operator, and $\alpha$ is a balancing hyperparameter. The first term updates the codewords to match the residuals, while the second aligns the residuals with the selected codewords. Additionally, a reconstruction loss $\mathcal{L}_{\text{rec}}$ ensures that the sum of quantized vectors, $\hat{E} = \sum_{n=1}^{N} v_{c_n}^n$, remains a faithful representation of the original vector $E$:

$$\mathcal{L}_{\text{rec}} = \|E - \text{MLP}(\hat{E})\|_2^2. \tag{3}$$

The overall training objective is $\mathcal{L} = \mathcal{L}_{\text{rec}} + \mathcal{L}_{\text{RQ}}$.

In contrast to to previous codebook training methods (Chen et al., 2025) using small, single-city data (often containing only a few thousand locations), training on our large-scale, multi-city dataset of millions of locations enables the model to learn a robust and general mapping from abstract semantic concepts to concrete tokens. This process results in a transferable, general vocabulary, laying the foundation for the model's generalization capabilities.

### 3.1.2 CODEBOOK ALIGNMENT WITH LLM

To integrate the new, semantically ungrounded tokens from our geographic codebook into the LLM, we propose a two-stage alignment methodology. This process first optimizes the static embeddings of the tokens and subsequently fine-tunes the LLM to comprehend their contextual usage.

**Stage 1: Optimizing Initial Token Embeddings.** To avoid a semantically void random initialization, we first set the initial embedding of each new token, $e_t^{(0)}$, to the mean of its constituent subword embeddings from the LLM's vocabulary. However, this serves only as a coarse approximation. To refine it, we formulate a composite loss function that aligns the new token embeddings with their original semantic space. For a given location ID, represented by the codeword sequence $\{t_{i_1}, \ldots, t_{i_k}\}$, we compute their average embedding $z$ and project it via a linear layer to obtain $\hat{y} = Linear(z)$. The alignment is optimized with the following loss:

$$\mathcal{L}_{\text{align}} = \mathcal{L}_{\text{main}} + \lambda_{\text{prior}} \mathcal{L}_{\text{prior}} + \lambda_{\text{coh}} \mathcal{L}_{\text{coh}}. \tag{4}$$

Here, $\mathcal{L}_{\text{main}}$ is a cosine similarity loss that aligns the projected embedding $\hat{y}$ with the original pre-quantization semantic vector $y$. This loss is regularized by two terms: $\mathcal{L}_{\text{prior}}$ maintains stability by

penalizing deviation from the initial embeddings, and $\mathcal{L}_{\text{coh}}$ leverages Pointwise Mutual Information (PMI) (Church & Hanks, 1990) to enforce similar representations for geographically co-occurring locations.

$$\mathcal{L}_{\text{main}} = \mathbb{E}\big[\max(0, 1-\cos(\hat{y}, y))\big], \mathcal{L}_{\text{prior}} = \frac{1}{M}\sum_{t\in\mathcal{N}}\|e_t - e_t^{(0)}\|_2^2, \mathcal{L}_{\text{coh}} = \frac{1}{|\mathcal{E}|}\sum_{(t,u)}\text{PMI}(t, u)\,\|e_t - e_u\|_2^2.$$
(5)

Upon completion, the optimized embeddings are integrated into the LLM's vocabulary matrix, establishing a robust semantic foundation for the next stage.

**Stage 2: Contextual Fine-tuning via Bidirectional Instruction-Tuning.** With semantically meaningful embeddings established, we fine-tune the LLM through a supervised, bidirectional instruction-tuning task, designed to enable it to understand and apply these tokens in context. The process has two complementary objectives:

1. **Interpretation (ID-to-Description):** Given a Location ID, the model is trained to generate its corresponding geographic description. This enables the LLM to interpret the semantics of specialized tokens.
2. **Retrieval (Description-to-ID):** Conversely, given a geographic description, the model must generate the correct Location ID. This enables the LLM to retrieve and apply the symbolic tokens as needed.

This bidirectional training ensures the model can proficiently map between symbolic identifiers and natural language, bridging the final gap. Detailed prompt designs are provided in Appendix D.

### 3.2 Mobility Understanding from Description to Summarization

While the semantic encoding in Section 3.2 provides the LLM with a mobility "vocabulary", genuine comprehension requires mastering the "grammar" of human movement—the ability to infer underlying spatiotemporal patterns from an MFM's latent trajectory sequence. To install this capability, we introduce a mobility-aware alignment curriculum. As illustrated in Figure 1(b), this strategy systematically guides the LLM from perceiving factual events to reasoning about abstract patterns. The curriculum unfolds in two progressive stages:

1. **Low-level mobility trajectory description task**: The LLM translates the MFM's latent sequence into a trajectory description of facts (e.g., "At time t, the user visited location l.")
2. **High-level spatiotemporal pattern summarization task**: The LLM learns to reason about the sequence encoding to infer abstract travel patterns, such as identifying frequently visited locations and modeling the temporal evolution of movement probabilities.

Crucially, this curriculum is not merely a pre-training understanding phase; it is architecturally integrated into the model's decision-making process for downstream applications. We formulate prediction and generation as a conditional, multi-part objective where the LLM is prompted to first output the high-level spatiotemporal feature summary before providing the final prediction or generation. This design choice is critical: it establishes a coherent **"understanding → prediction | generation" reasoning chain**. By forcing the model to articulate its reasoning first, we provide a strong inductive bias that compels it to base its predictions on inferred spatiotemporal patterns rather than on superficial sequence correlations. Detailed prompt designs are provided in the Appendix D.

**Training loss**. The aforementioned tasks of understanding, prediction, and generation are optimized through supervised fine-tuning. Given an input trajectory sequence $X_{\text{seq}}$, it is first processed by the MFM encoder, denoted as $g_\phi$. The resulting representation is then projected into the LLM's input embedding space via a lightweight MLP, yielding the final conditioning hidden state $H_{\text{seq}} = \text{MLP}(g_\phi(X_{\text{seq}}))$. Conditioned on this mobility representation $H_{\text{seq}}$ and a corresponding text instruction $X_{\text{Ins}}$, the LLM is trained to autoregressively generate the target text output $y = (y_1, y_2, \ldots, y_N)$. The model's parameters $\theta$ are optimized by minimizing the negative log-likelihood of the ground-truth sequence. The loss function $\mathcal{L}$ is defined as:

$$\mathcal{L} = -\frac{1}{N}\sum_{t=1}^{N}\log P_\theta(y_t \mid y_{<t}, X_{\text{Ins}}, X_{\text{seq}})$$
(6)

## 3.3 Interactive Mobility Generation

While mobility foundational models excel at capturing historical patterns, their architecture inherently lacks the flexibility to generate trajectories under arbitrary, open-ended scenarios. Integrating LLMs offers a powerful new avenue to address this limitation, enabling the generation of trajectories that conform to diverse, language-specified conditions. The core challenge of this task lies in the dual objective of strictly adhering to the scenario's explicit spatiotemporal constraints while maintaining high fidelity to the user's ingrained behavioral patterns. To resolve this tension, we introduce a Self-Reflective Reasoning strategy, which begins with a baseline generated trajectory derived from user history and applies the minimal necessary edits to satisfy the new constraints, ensuring the final trajectory is both scenario-compliant and behaviorally consistent.

### 3.3.1 Self-Reflective Reasoning

Our **Self-Reflective Reasoning** operationalizes the "minimal edits" principle through a structured, iterative process, as illustrated in Figure 1(c). Instead of generating a trajectory in a single pass, the model engages in a deterministic loop of generation, critique, and refinement. The process unfolds as follows:

1. **Baseline Generation.** The model first generates an initial future trajectory based on the user's historical data. This serves as a critical "zero-scenario" baseline, a starting point that is by definition fully consistent with the user's established spatiotemporal patterns.
2. **Iterative Refinement.** The model then enters a refinement loop. It compares the current trajectory against the explicit spatiotemporal constraints of the target scenario. If any statistical mismatches are detected, the model proposes a targeted edit. After applying the edit, the modified trajectory is re-evaluated.
3. **Termination.** This loop continues until the trajectory fully satisfies all scenario constraints. Upon reaching this self-consistent state, the model outputs the final edited trajectory, along with a structured summary of the edits and their justifications.

We address the dual objectives outlined previously through a simple yet powerful heuristic: explicitly instructing the model to seek a solution requiring the fewest number of edits, ensuring that the final output is a true synthesis, rather than a completely new, unrelated behavior. To guide the model in planning edits, we define a discrete action space containing three permissible edit operations: (i) adding a trajectory point, (ii) deleting a trajectory point, or (iii) modifying the time and/or position of an existing point.

### 3.3.2 Reward Modeling for RL Training

We implement iterative trajectory reasoning using Group Relative Policy Optimization (GRPO) (Shao et al., 2024b). Compared with PPO (Schulman et al., 2017), GRPO eliminates the need for a separately trained value function by using group-relative rewards to compute advantages, significantly reducing memory and computational overhead, which better suits LLM tasks. The model is trained to follow the reasoning template detailed in Table 1.

Table 1: Template for Self-Reflective Reasoning with GRPO

| |
|---|
| Please answer the following questions step by step. You need to think and reason before answering, outputting your reasoning process between `<think>` and `</think>`, and providing your final answer between `<answer>` and `</answer>`. |
| Input: Historical trajectory data, initial generated trajectory, spatiotemporal constraints. |
| Task: Modify the initial trajectory data based on the historical data and the spatiotemporal constraints of the scene. Ensure that the modified trajectory conforms to the given statistical spatiotemporal characteristics and uses the minimum modification step size. |

The design of the reward function is guided by a crucial objective: **distributional consistency**. Unlike common reasoning tasks that target an exact-match (EM) solution, our goal is to generate trajectories that align with the correct statistical distribution. Consequently, we formulate a reward function based on matching key spatiotemporal statistical properties (e.g., travel probability at different time periods, and probability distribution of visited places) rather than the ground-truth

trajectory. Let $\phi(\tau)$ denote the statistical feature of trajectory $\tau$. The reward is:

$$R_{\text{distribution}}(\tau) = \sum_{k=1}^{K} \mathbf{1}[\phi_k(\tau) = \phi_k(\tau^*)],\qquad(7)$$

where $\tau^*$ is the ground truth trajectory. Each matched feature contributes $+1$ for the reward. Additionally, to avoid unrealistic length deviations, we penalize discrepancies between generated and ground-truth lengths:

$$R_{\text{length}}(\tau) = -\frac{|\,|\tau| - |\tau^*|\,|}{|\tau^*|}.\qquad(8)$$

In summary, the total reward can be expressed as follows: $R(\tau) = R_{\text{distribution}}(\tau) + R_{\text{length}}(\tau)$.

In addition, to further ensure training stability, we begin with supervised fine-tuning (SFT) as a cold-start phase prior to GRPO training. This initialization allows the model to produce well-structured outputs and prevents instability during early reinforcement learning. Therefore, we do not need the format-based rewards (e.g., validating the `<think>` and `<answer>` tags), as SFT sufficiently enforces adherence to the training template.

## 4 EXPERIMENT

### 4.1 EXPERIMENTAL SETUP

**Dataset**: We evaluated our approach on four real-world human mobility datasets (Atlanta, Chicago, Seattle, and Washington, D.C., USA). The geographic space of each city was discretized into 500-meter grid cells, with a minimum temporal granularity of 30 minutes. User trajectories were constructed using a sliding window covering three consecutive days, and trajectories with fewer than five trips were discarded to reduce sparsity and noise. And if the trajectory exceeded 145 points, only the most recent 145 trajectory points were retained For each visited location, we combined geographic coordinates with semantic information (e.g., points of interest) extracted from OpenStreetMap. We take careful measures to ensure that ethical considerations are fully addressed in the use of data. Further details on the dataset statistics and preprocessing are provided in the Appendix C.

**Evaluation Metrics**: For **prediction task**, we adopt commonly used metrics, hating rating ($HR@1$) to evaluate the prediction performance (Han et al., 2025; Chen et al., 2025). For **generation task**, we adopt commonly used metrics $BLEU$, $TVD$, and $JSD$ to measure the time and location similarity between the generated sequence and the real sequence respectively (Reed et al., 2016; Wang et al., 2024). For the more details on all metrics above, please refer to the Appendix E.

**Baselines**: For **prediction task**, we selected DeepMove (Feng et al., 2018), TrajBert (Si et al., 2023), GETNext (Yang et al., 2022), TrajFM (Lin et al., 2024), Unitraj (Zhu et al., 2024b), and Traj-MoE (Han et al., 2025) as traditional deep learning approaches. Among these, Unitraj and TrajMoE are pre-trained foundation sequence methods. For LLM-based prediction approaches, we selected Mobility-LLM (Gong et al., 2024) and QT-Mob (Chen et al., 2025). For **generation task**, we selected two recent diffusion-based approaches, DiffTraj (Zhu et al., 2023) and Marionette (Deng et al., 2025) and LLM-enhanced generation approaches, COPB (Shao et al., 2024a) and LLMob (Wang et al., 2024). For more details on the above baselines, see the Appendix F.

**Implementation Details**: The experiments were conducted on four NVIDIA A800 40G GPUs, using Qwen2.5-7B (Hui et al., 2024) as the backbone network and TrajMOE (Han et al., 2025) as the enhanced mobility foundation model. We employed LoRA fine-tuning (Hu et al., 2022) and parallel training for acceleration. For the reflective reasoning experiments, we utilized two additional NVIDIA A100 80G GPUs with Qwen3-4B (Yang et al., 2025) as the backbone. For more experimental details, please refer to the Appendix G.

### 4.2 MOBILITY PREDICTION

**Next Location Prediction**: We evaluated the performance of all methods on four benchmark datasets. Note that methods supporting cross-city pre-training (e.g., TrajMoE) were trained on a

Table 2: Experiment result on prediction task(HR@1).

| | DeepMove | GETNext | TrajFM | Unitraj | TrajMoE | Mobility-LLM | QT-Mob | **MoveFM-R** | Improve |
|---|---|---|---|---|---|---|---|---|---|
| Atlanta | 0.171 | 0.178 | 0.196 | 0.210 | 0.245 | 0.214 | 0.240 | **0.281** | +14.7% |
| Chicago | 0.188 | 0.189 | 0.212 | 0.219 | 0.269 | 0.218 | 0.306 | **0.334** | +9.2% |
| Seattle | 0.220 | 0.227 | 0.255 | 0.283 | 0.309 | 0.270 | 0.315 | **0.368** | +16.8% |
| Washington | 0.204 | 0.197 | 0.202 | 0.215 | 0.265 | 0.224 | 0.286 | **0.328** | +14.7% |

Table 3: Experiment result on zero-shot and few-shot(HR@1).

| Method | Atlanta | | Chicago | | Seattle | | Washington | |
|---|---|---|---|---|---|---|---|---|
| | zero-shot | few-shot | zero-shot | few-shot | zero-shot | few-shot | zero-shot | few-shot |
| TrajMoE | 0.121 | 0.151 | 0.085 | 0.098 | 0.146 | 0.194 | 0.141 | 0.168 |
| QT-Mob | 0.132 | 0.203 | 0.242 | 0.255 | 0.218 | 0.244 | 0.242 | 0.271 |
| Ours | **0.164** | **0.264** | **0.280** | **0.309** | **0.262** | **0.294** | **0.272** | **0.292** |
| Improve | +24.24% | +30.05% | +15.70% | +21.18% | +20.18% | +20.49% | +12.40% | +7.75% |

mixed dataset from all four cities and tested on each city's dataset to maximize the benefits of their pre-training. The results, summarized in Table 2, reveal several key observations: First, our method improves prediction accuracy by over 10% on average across all datasets. And compared to TrajMoE (selected as the fundamental model for our method), our method, achieves over 20% improvement, demonstrating its ability to enhance pre-trained fundamental models. Moreover, our approach outperforms the LLM baseline, which relies solely on plain text input, by an additional 10%, emphasizing the value of spatiotemporal features captured by domain-specific models.

**Zero-Shot and Few-Shot Performance** For the zero-shot experiments, we pre-trained the model on data from three cities and tested it on the remaining cities (treated as novel environments). For the few-shot experiments, we fine-tuned the model on 500 examples from the remaining cities and then tested it. The results, presented in Table 3, reveal several key findings. First, our approach consistently outperforms both the strongest sequence-based and LLM baseline models (TrajMoE,QT-Mob) in terms of zero-shot and few-shot performance across all four cities, demonstrating robust generalization to novel environments. Second, the LLM-based approach, QT-MOB, comprehensively outperforms the purely sequence-based model, TrajMoE, highlighting the impressive ability of language models to transfer knowledge across diverse urban environments. Notably, our approach achieves zero-shot accuracy in all four cities that surpasses the classic method, DeepMove, even when the latter is fine-tuned on the full dataset, further emphasizing the strong generalization capabilities of our method.

## 4.3 MOBILITY GENERATION

**Unconditional Generation** For generation tasks, we focus more on the distribution consistency (fidelity) between the generated sequences and the real sequences rather than accuracy. We evaluated all methods on four city datasets, where the task was

Table 4: Performance of unconditional generation.

| Method | Time | | | Location | | |
|---|---|---|---|---|---|---|
| | Bleu ↑ | TVD ↓ | JSD ↓ | Bleu ↑ | TVD ↓ | JSD ↓ |
| DiffTraj | 0.387 | 0.117 | 0.009 | 0.076 | 0.494 | 0.220 |
| Marionette | 0.582 | 0.082 | 0.008 | 0.092 | 0.346 | 0.102 |
| COPB | 0.426 | 0.096 | 0.009 | 0.084 | 0.382 | 0.133 |
| LLMob | 0.605 | 0.085 | 0.007 | 0.095 | 0.323 | 0.095 |
| **Ours** | **0.628** | **0.064** | **0.006** | **0.136** | **0.250** | **0.062** |

to generate a user's trajectory on the third day based solely on historical data from the previous two days. The results, presented in Table 4, reveal several key observations. For more details on indicator calculations, please refer to the appendix E.

First, our method achieves state-of-the-art performance across all metrics($BLEU$, $TVD$, and $JSD$) for both temporal and location distribution. Second, by leveraging the fundamental mobility model's capacity to extract informative features from numerical sequences, our method significantly outperforms all LLM-based baselines(COPB,LLMob), which highlights the importance of grounding LLM reasoning in domain-specific representations rather than relying exclusively on textual input. Furthermore, our method surpasses pure sequence modeling approaches(DiffTraj,Marionette) by benefiting from the semantic understanding and reasoning capabilities of LLMs. Together, these findings demonstrate that integrating structured trajectory features with LLM provides consistent advantages over both traditional architectures and LLM-only methods.

**Conditional Generation** We evaluated the model's conditional trajectory generation performance in three representative scenarios: (i) **late-night commuters**, where nighttime trips account for over three-quarters of all trips; (ii) **users with temporary travel plans**, such as those making a last-minute decision to visit or not visit a place; and (iii) **weekend**

Table 5: Performance of conditional generation.

| Method | Time | | | Location | | |
|---|---|---|---|---|---|---|
| | Bleu ↑ | TVD ↓ | JSD ↓ | Bleu ↑ | TVD ↓ | JSD ↓ |
| w/o SR | 0.433 | 0.186 | 0.032 | 0.044 | 0.662 | 0.412 |
| Scenario-i | **0.455** | **0.156** | **0.023** | **0.091** | **0.545** | **0.321** |
| w/o SR | **0.532** | **0.109** | **0.010** | 0.128 | 0.339 | 0.124 |
| Scenario-ii | 0.506 | 0.121 | 0.011 | **0.148** | **0.243** | **0.080** |
| w/o SR | **0.414** | **0.153** | **0.019** | 0.080 | 0.560 | 0.323 |
| Scenario-iii | 0.395 | 0.167 | **0.019** | **0.085** | **0.443** | **0.238** |

**users,** where historical sequences correspond to Thursdays and Fridays, and generated trajectories correspond to Saturdays. These scenarios capture diverse travel patterns and provide a comprehensive test of scenario-based generation. Detailed division information is available in appendix I.

The results (shown in Table 5) show that our approach achieves significant improvements over scenario-free generation (represented as 'w/o SR') in most scenarios, though the temporal distributions for users with explicit travel plans and weekend users are slightly inferior. This success stems from our self-reflective reasoning, which effectively exploits scenario-specific spatiotemporal constraints. For example, it enforces temporal regularity for late-night commuters (improving time) and uses the destination as a strong spatial anchor for users with explicit plans (improving space). Conversely, the slight temporal decline reveals a challenge with high stochasticity and pattern shift; the model struggles to predict highly variable weekend timing from weekday data or when travel times are inherently random despite a fixed destination. In such cases, the unconditional model's more generalized distribution proves advantageous.

## 4.4 ABLATION STUDY

To validate the effectiveness of each component in our framework, we conducted ablation studies on four datasets. We evaluated the model under four settings: (i) without CB (codebook), (ii) without RU (representation understanding), (iii) without FM (base model). The results for the prediction task are in Table 6, and for the generation task in Table 7. Key observations include:

First, removing both the base model and the codebook results in a significant drop in performance, highlighting the importance of the spatiotemporal trajectory features and spatial semantics provided by the base model and the structured position encoding. Second, removing representation understanding results in a moderately consistent drop in performance on both tasks, highlighting that fine-grained feature understanding helps the LLM better exploit spatiotemporal information. This effect is slightly more pronounced in the generation task. Overall, these ablation results confirm that each component makes a meaningful contribution and that they collectively enhance trajectory prediction and generation.

Table 6: Results of ablation studies (prediction).

| Method | Atlanta | Chicago | Seattle | Washington |
|---|---|---|---|---|
| Ours | 0.281 | 0.334 | 0.368 | 0.328 |
| w/o CB | 0.243 | 0.310 | 0.326 | 0.306 |
| w/o RU | 0.270 | 0.328 | 0.350 | 0.314 |
| w/o FM | 0.259 | 0.318 | 0.337 | 0.304 |

Table 7: Results of ablation studies (generation).

| Method | Time | | | Location | | |
|---|---|---|---|---|---|---|
| | Bleu ↑ | TVD ↓ | JSD ↓ | Bleu ↑ | TVD ↓ | JSD ↓ |
| Ours | 0.628 | 0.064 | 0.006 | 0.136 | 0.250 | 0.062 |
| w/o CB | 0.598 | 0.090 | 0.007 | 0.112 | 0.273 | 0.072 |
| w/o RU | 0.613 | 0.072 | 0.006 | 0.108 | 0.265 | 0.068 |
| w/o FM | 0.594 | 0.087 | 0.007 | 0.108 | 0.278 | 0.074 |

## 5 CONCLUSION

This research repositions the ultimate goal of human mobility modeling: moving beyond mere pattern prediction to achieve a genuine understanding of human intent. Our work demonstrates that the key to this evolution lies in the thoughtful synthesis of statistically powerful MFMs and the deep semantic reasoning of LLMs. We have shown that this synergy is not just a theoretical possibility but a practical reality, creating models that can interpret the "why" behind the "where". The value of this new paradigm is profound. It unlocks the ability to interact with and steer mobility generation through natural language, making sophisticated simulation and analysis accessible to a broader range of experts, including urban planners and social scientists.

ETHICS STATEMENT

We have implemented robust measures to ensure the ethical handling of data throughout this study, with a focus on privacy, security, and bias mitigation. To protect individual privacy, the trajectory data underwent a rigorous anonymization process and contains no personally identifiable information (PII). To further render the re-identification of individuals infeasible, random noise was added to all location data points, a technique known as location perturbation. All datasets are stored on secure, encrypted servers with strict access control protocols, limiting access to authorized research personnel bound by non-disclosure agreements. Furthermore, to proactively address fairness, the dataset intentionally excludes any demographic or user-specific attributes, such as gender, race, or age. This design inherently mitigates the risk of our model learning or perpetuating societal biases related to these characteristics. We believe this research holds the potential for significant positive societal impact by contributing to a deeper understanding of human mobility for applications in areas like intelligent urban planning and transportation systems.

REPRODUCIBILITY STATEMENT

To ensure the reproducibility of our research, we commit to making our work as transparent and accessible as possible.

- **Code:** The source code for our proposed model, experimental setup, and evaluation scripts will be made publicly available in a GitHub repository upon publication of this work. The repository will include detailed instructions for setting up the environment and running the experiments.

- **Implementation Details:** Key hyperparameters and architectural choices for our model are described in the main paper. A comprehensive list of all hyperparameters, along with details about the computational environment (hardware, software libraries, and versions), will be provided in the `README.md` file of our code repository.

The implementation of MoveFM-R is available online at `https://anonymous.4open.scie nce/r/MoveFM-R-CDE7/`

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

## A    USE OF LLMS

We used LLMs to assist in writing the paper, such as identifying typos and correcting grammatical errors, as well as polishing some paragraphs.

## B    SEMANTIC INFORMATION DESCRIPTION

**Semantic Information Example**

- **Location Address:** The location is situated at South Street, Hapeville, 30354, United States.
- **Geographic Coordinates and Boundary:** The center of the location is at **latitude 33.6544382** and **longitude -84.4045157**. The area is bounded by:
  - **Minimum latitude:** 33.654528
  - **Maximum latitude:** 33.6548927
  - **Minimum longitude:** -84.403952
  - **Maximum longitude:** -84.4036685
- **OpenStreetMap (OSM) Details:**
  - **OSM Type:** way
  - **OSM ID:** 975678110
  - **Place ID:** 132886
- **Points of Interest (POIs):** The location includes 1 fast food, 1 restaurant.

**POI Categories**: "gid", "finance", "public", "transport", "entertainment", "health", "service", "education", "government", "religion", "accommodation", "food", "cafe", "fast_food", "ice_cream", "pub", "restaurant", "shop_beauty", "shop_clothes", "boutique", "shop_transport", "retail", "commodity", "marketplace", "home-improvement", "sport", "public_transport", "kindergarten", "office", "recycling", "travel_agency", "tourism", "shop_livelihood", "residential", "dormitory".

## C    DATASET DETAILS

**Dataset Statistics** The statistical overview of the datasets used is presented in Table 8.

Table 8: Statistical information for the trajectory datasets used in our experiments.

| City | Duration | Locations | Trajectories |
|------|----------|-----------|--------------|
| Atlanta | 7 days | 1,175 | 200,000 |
| Chicago | 7 days | 4,166 | 200,000 |
| Seattle | 7 days | 1,046 | 200,000 |
| Washington | 7 days | 1,361 | 200,000 |

## D    TASK PROMPT EXAMPLES

**Geographic Location Understanding**:

- **loc2id:** Your task is to infer the corresponding Location index based on the geographic location information: [location]\n Its Location index is :

- **id2loc:** Your goal is to learn and remember the geographic location information represented by the Location index.\n The geographic information of Location index [index] is :

**Understanding + Prediction**:

> This is a user trajectory prediction task. Your goal is to predict the next location index using both an authoritative trajectory text and a possibly noisy sequence embedding.
>
> **Provided:**
> - Ground-truth trajectory text (always correct): `<traj_data>`
> - Sequence embedding of the trajectory (auxiliary signal): `<sequence>`
>
> **Conflict/irrelevance handling:**
> - If any embedding-based interpretation contradicts the trajectory text or reflects a trajectory largely unrelated to the text, disregard the embedding interpretation and rely on the text.
> - Only incorporate embedding cues that align with the text.
>
> **Tasks:**
> 1. Based on the trajectory text and your analysis of the sequence embedding (ignore it if inconsistent with the text), produce the user's spatio-temporal trajectory features, filling the template exactly:
>
> *Summary of the spatio-temporal trajectory features:*
>
> *- Most frequently visited locations (visited more than once): [Output at most the first three (if any)]*
>
> *- Probability of visits by time period (rounded to 5%): [list all periods with probability values, even if 0%]*
>
> 2. Using these features and the inputs(if sequence embedding appears inconsistent with the textual trajectory, ignore it), predict the user's next location index.
>
> Output only the completed feature block and the final prediction. Do not include explanations.

**Understanding + Generation**:

> The user's original trajectory data contains weekday, timestamp, and location index information. Below is the encoded vector of the user's trajectory sequence for the past two days:
>
> `<sequence>`
>
> In addition, there also has a special text format description of the user's historical trajectory as supplementary information: `<history_text>`.
>
> You need to first carefully interpret both the encoded trajectory sequence (embedding) and the historical textual trajectory description, and then complete the following two tasks:

**Step 1:** Generate 'Summary of the trajectory preferences for this user' strictly in the following format:

*Summary of the trajectory preferences for this user:*

*- Most frequently visited locations (visited more than once): [Output at most the first three (if any)]*

*- Probability of visits by time period (rounded to 5%): [list all periods with probability values, even if 0%]*

*- Frequently visited locations during each time period: [list per period; if none, explicitly say 'No location was visited more than once'].*

**Step 2:** Based on both the summary and the encoded vector together with the historical textual trajectory description, generate the user's trajectory activity for the next day. Each data point in the generated trajectory should be in the format: *At [time], visited location [location index].*

SELF-REFLECTION

You are an intelligent assistant skilled at asking questions and thinking. Please solve the following problem step by step. First, you should think through the reasoning process and then provide the answer to the user. The reasoning process and answer are contained in the `<think> </think>` and `<answer> </answer>` tags, respectively, i.e., `<think>reasoning process here </think><answer>answer here </answer>`.

You need to complete the following trajectory modification task:

**Input:**

Completely known input:

1. Given two days of historical behavior data

2. Previously generated user trajectory data for the next day

3. Statistical spatiotemporal features of historical behavior data

4. Statistical spatiotemporal features of real data for the next day

5. Given Modification Steps: `[constraint]`, and then K trajectory modifications (the specific value of K is determined by your own analysis).

**Task Requirements:** Based on fully known inputs, modify and improve previously generated trajectory data for the next day, using the given modification steps, and ensure that the modified trajectory data is maximally consistent with the Statistical spatiotemporal features of real data for the next day. The analytical support should only be derived from fully known inputs.The final output should include a summary of the modification steps and the corresponding reasons, as well as the final user trajectory for the next day after the modification steps. Be careful not to analyze `<a_x><b_x><b_x><d_x>` separately. `<a_x><b_x><b_x><d_x>` together form a whole to describe a specific location. Do not add or generate new `<a_x><b_x><b_x><d_x>` when modifying. When modifying a previous future trajectory, only locations that have appeared in history and previously generated future trajectories, as well as locations that have appeared in the spatiotemporal features corresponding to the given future day's real trajectory data, can be used. For the time modification, you can generate timestamps that are not in the historical sequence or previously generated future tracks.Note that deleting a track, adding a track, or modifying a track (either location, time, or both) is considered a single operation. Please complete the reasoning analysis based on this,using as few modification steps as possible.

**Specific input data is as follows:**

Fully known input:

1. Given historical behavior data: `[data1]`

2. Previously generated user trajectory data for the next day: `[data2]`

3. Statistical spatiotemporal features of historical behavior data: `[data3]`

4. Statistical spatiotemporal features of real data for the next day: `[data4]`

# E    EVALUATION METRICS

### PREDICTION TASK

The Hit Rate (or Accuracy) measures the proportion of correctly predicted next locations within the top-$k$ recommendations. The formula is:

$$\text{Hit Rate@k} = \frac{1}{|U|} \sum_{u \in U} \mathbb{I}(\text{rank}_u \leq k) \tag{9}$$

where $|U|$ is the total number of users, and $\mathbb{I}(\cdot)$ is an indicator function that is 1 if the true next location is within the top-$k$ predictions, and 0 otherwise.

### GENERATION TASK

**Bilingual Evaluation Understudy (BLEU):**    A metric for evaluating the quality of generated text against a reference.

$$\text{BLEU} = \text{BP} \cdot \exp \left( \sum_{n=1}^{N} w_n \log p_n \right) \tag{10}$$

where $\text{BP} = \min\left(1, e^{1-r/c}\right)$ is the brevity penalty, $p_n$ is the modified $n$-gram precision, $r$ is the reference length, and $c$ is the candidate length.

**Total Variation Distance (TVD):**    Measures the distance between two probability distributions.

$$\text{TVD}(P, Q) = \frac{1}{2} \sum_{i=1}^{k} |P(i) - Q(i)| \tag{11}$$

where $P$ and $Q$ are probability distributions over $k$ classes, $P(i)$ is the predicted probability of class $i$, and $Q(i)$ is the ground truth probability.

**Jensen-Shannon Divergence (JSD):**    A smoothed and symmetric measure of the similarity between two probability distributions.

$$\text{JSD}(P\|Q) = \sqrt{\frac{1}{2} D_{\text{KL}}(P\|M) + \frac{1}{2} D_{\text{KL}}(Q\|M)} \tag{12}$$

where $M = \frac{1}{2}(P + Q)$ is the midpoint distribution, and $D_{\text{KL}}$ is the Kullback-Leibler divergence:

$$D_{\text{KL}}(P\|Q) = \sum_{i=1}^{k} P(i) \log \frac{P(i)}{Q(i)} \tag{13}$$

### GENERATING INDICATOR ALGORITHMS

Our time data is granular with half-hourly intervals. We calculate JSD in half-hourly buckets, while TVD and BELU are implemented using standard algorithm libraries such as scipy and nltk.

# F    BASELINE DETAILS

Our baseline selection spans different methodological families to ensure a comprehensive evaluation. Below is a brief introduction to the core principle of each selected model.

### PREDICTION BASELINES

- **DeepMove** (Feng et al., 2018) is an attentional recurrent neural network that captures both long-term periodic patterns and short-term sequential regularities in user mobility.
- **TrajBert** (Si et al., 2023) adapts the powerful BERT architecture to model trajectories by treating locations as tokens and learning deep, bidirectional contextual representations for prediction.

- **GETNext** (Yang et al., 2022) integrates a graph neural network to explicitly learn spatial relationships between locations with a Transformer-based encoder to capture complex spatio-temporal dependencies.

- **TrajFM** (Lin et al., 2024) is a foundation model for trajectories that is pre-trained on a massive dataset to learn universal mobility patterns adaptable to various downstream tasks.

- **Unitraj** (Zhu et al., 2024b) is a universal pre-trained model that unifies the representation of diverse trajectory data types, including spatio-temporal points, semantic texts, and graph structures.

- **TrajMoE** (Han et al., 2025) employs a Mixture-of-Experts (MoE) architecture where different "expert" sub-networks specialize in modeling distinct mobility patterns for more accurate and robust predictions.

- **Mobility-LLM** (Gong et al., 2024) is a large language model-based framework that reformulates trajectory prediction as a language modeling task by converting mobility data into textual sequences.

- **QT-Mob** (Chen et al., 2025) enhances LLMs for mobility prediction by incorporating a query-time adaptation mechanism that retrieves and integrates relevant external spatio-temporal knowledge at the time of inference.

GENERATION BASELINES

- **DiffTraj** (Zhu et al., 2023) applies a denoising diffusion probabilistic model to generate realistic and diverse human trajectories by progressively refining a random noise signal into a structured sequence.

- **Marionette** (Deng et al., 2025) is a controllable trajectory generation model based on guided diffusion, allowing for the synthesis of trajectories that adhere to specific user-defined constraints or conditions.

- **COPB** (Shao et al., 2024a) leverages the Chain-of-Thought prompting technique with large language models to iteratively reason about user preferences and construct plausible, context-aware trajectories.

- **LLMob** (Wang et al., 2024) is a comprehensive framework that utilizes the generative and reasoning capabilities of large language models to produce human-like trajectories based on user profiles and historical data.

## G  IMPLEMENTATION DETAILS

This experiment used four NVIDIA A800 40GB GPUs. We chose Qwen2.5-7B (Hui et al., 2024) as the backbone network. The experiments used the AdamW optimizer, with a cosine annealing learning rate and a warmup ratio of 0.03. The maximum learning rate for the cosine annealing algorithm was set to 1e-4, and both the minimum warmup learning rate and the initial warmup learning rate were set to 2e-5. We performed LoRA (Hu et al., 2022) fine-tuning and parallel training acceleration. All experiments were conducted with a maximum of 5 training epochs and a batch size of 96, and the best-performing model on the validation set was selected for testing. For the reflective inference experiments, we additionally used two NVIDIA A100 80G GPUs and chose Qwen3-4B (Yang et al., 2025) as the backbone network. Due to limited computing resources, we fixed the random seed to 42 and ran the experiment only once. For information on the model parameters involved in the method, please refer to the appendix below H.

## H  MODEL CONFIGURATIONS

**Codebook Model**: The trajectory discretization is performed by a vector quantization model. Its encoder is an MLP with hidden layer dimensions of $[2048, 1024, 512, 256, 128, 64]$. The model utilizes four separate codebooks, each containing 512 embeddings of 64 dimensions. For training, we used the AdamW optimizer with a learning rate of $1 \times 10^{-3}$ and a batch size of 1024.

**Mobility Foundation Model**: Our mobility foundation model is a Transformer-based architecture. It is configured with 4 layers, 4 attention heads, and an embedding dimension of 512. The model

was trained for 50 epochs using a learning rate of $3 \times 10^{-4}$ and a batch size of 8 to process trajectory sequences with a maximum length of 145.

**Large Language Model Fine-Tuning**: For the supervised fine-tuning (SFT) phase, we employed the **Qwen2.5-7B** model. Due to computational constraints during the subsequent Generative Rejective Policy Optimization (GRPO) stage, we trained an auxiliary **Qwen3-4B** model. This smaller model was tasked with the self-reflection and reasoning steps, enabling us to effectively complete the GRPO training on the primary 7B model within our resource limits.

## I    CROWD FILTERING CRITERIA

**Late-night Commuters**: We define **Late-night Commuters** as individuals who undertake trips between 10 PM and 6 AM. The specific criterion for this classification is that a user's trips within this time frame must account for more than three-quarters (75%) of their total daily trips. Trajectories belonging to this user group were specially flagged to analyze their distinct mobility patterns.

**Users with Temporary Travel Plans**: To isolate and analyze non-habitual or temporary travel behaviors, we established criteria to identify users with transient travel intentions. Our methodology involves an examination of the top three most frequently visited locations within a user's historical and future trajectories.

- **Identification of New Plans:** If a location that is not among a user's three most historically frequent locations appears in their future trajectory, we classify this as the user having made a new plan to visit a previously infrequently visited location.
- **Identification of Canceled Plans:** Conversely, if a location that ranks among the top three most visited places in a user's historical trajectory does not appear in their planned future trajectory, we infer that the user has canceled a previously planned visit to a frequented location.

**Weekend Users**: We constructed a specific data subset where the historical series contains trajectory data from Thursday and Friday, which is then used to predict the user's trajectory on Saturday. Consequently, only users with complete and valid trajectory data for the preceding two days were included in this predictive task.

## J    VISUALIZATION OF GENERATED TRAJECTORIES

We visualized the temporal and location distributions of trajectories generated by representative algorithms under unconditional generation. As shown in Figure 2, compared to the baseline, the trajectories generated by our method are much closer to the distribution of true trajectories. In particular, for location distribution, our method shows significant improvements in both high-frequency and long-tail regions, demonstrating a higher fidelity to real-world mobility patterns.

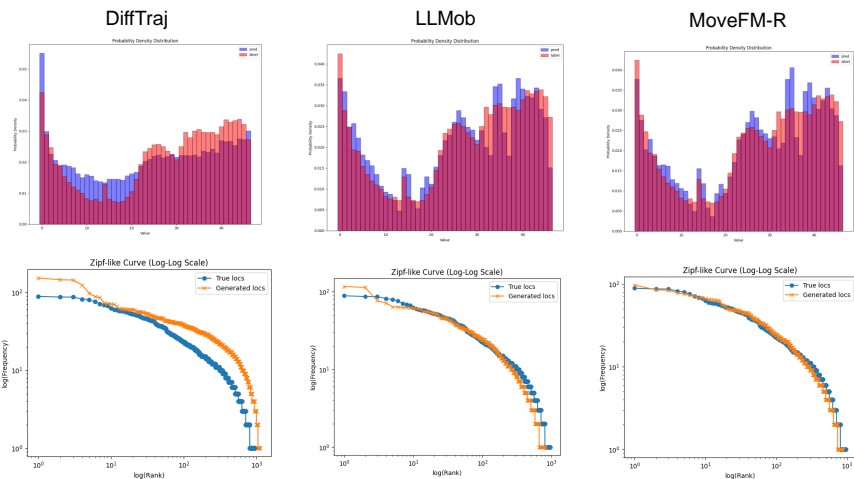

Figure 2: **Comparison of Temporal and Location Distributions.** We evaluate the distributions of generated trajectories from our model (MoveFM-R) against baselines (DiffTraj, LLMob). **Top row:** Visualization of the temporal distribution. The generated distribution (red) from our model more accurately matches the true temporal distribution (blue) of user activities over time. **Bottom row:** Visualization of the location distribution on a log-log scale (Zipf-like plot). The curve for our generated data (orange) shows a much tighter fit to the ground-truth data (blue) across the entire spectrum, from popular (head) to rare (tail) locations.

