# OpenReview forum: "MoveFM-R: Advancing Mobility Foundation Models via Language-driven Semantic Reasoning"
_ICLR.cc/2026/Conference — Submitted to ICLR 2026_

### Official Review · Reviewer_rBv8 · 2025-10-15

**Soundness:** 2
**Presentation:** 2
**Contribution:** 2
**Rating:** 4
**Confidence:** 2

**Summary:**

This study proposed two schemes to enhance the foundation model in human mobility tasks. First, they integrate the new geographical token into the LLM to enhance the semantic understanding of the LLM on the geospatial context. Second, they design two fine-tuning tasks for LLM to learn to reason about the mobility trajectory. They conduct multiple tasks in the domain of human mobility to demonstrate the performance of the proposed schemes.

**Strengths:**

1. The motivation of this study is meaningful.
2. Extensive experiments are conducted to validate the performance.

**Weaknesses:**

1. The paper suffers from weak organization; too many elements are combined within sections, which reduces overall coherence. It is recommended to focus on some key points instead of gathering all the related stuff.
2. Some of the claims lack support. For example, "LLMs inherently lack an understanding of raw geographic coordinates.". We can check gps coordinates in ChatGPT and find an exact location without searching.
3. Although extensive experiments are conducted, the proposed schemes involve numerous optimization processes, which may require costly sensitivity analyses. This complexity reduces their practicality and efficiency for real-world applications.

**Questions:**

1.  How do you balance the R_distribution and R_length?
2.  How can you measure whether the codebook is well aligned with the LLM?
3.  What is the necessity of each subloss in equation (4)?
4. Why only conduct a compasiron study for unconditional generation but not for conditional generation?

---

### Official Review · Reviewer_opDh · 2025-10-17

**Soundness:** 2
**Presentation:** 3
**Contribution:** 2
**Rating:** 2
**Confidence:** 3

**Summary:**

This paper introduces MoveFM-R, a novel framework that integrates Large Language Models to achieve a more comprehensive understanding of human mobility. First, it employs a semantically enhanced location encoding mechanism that transforms raw geographic coordinates into discrete, interpretable tokens. Second, a curriculum-based alignment process progressively trains the LLM to comprehend movement patterns. Third, an Interactive Mobility Generation module introduces a self-reflective reinforcement learning strategy based on GRPO, allowing the model to iteratively refine and generate trajectories that are both semantically meaningful and physically plausible. Extensive experiments show that MoveFM-R achieves state-of-the-art results in mobility prediction and generation.

**Strengths:**

The paper is well-motivated and clearly presented. From prospective of methodology, the curriculum-based alignment of MoveFM-R smartly align the LLM’s reasoning with mobility patterns. The self-reflective reinforcement learning component based on GRPO is innovative, showing a thoughtful integration of reasoning-based generation with mobility related problems. Empirically, the paper demonstrates strong performance. Evaluations span multiple real-world urban datasets show the superiority of MoveFM-R

**Weaknesses:**

* W1: The proposed semantic codebook component (Section 3.1.1) appears conceptually similar to the quantized trajectory encoding used in QT-Mob and [1]. It's difficult to consider this module a core innovation.

[1] Generative Next POI Recommendation with Semantic ID

* W2:  The alignment stage introduces three losses, but their  necessity are not clearly justified. For example, if tokens are represented as abstract codebooks (e.g., a_x  b_x  c_x  d_x), enforcing consistency with their initial meaningless embeddings may not be semantically meaningful.  What's more, LLM often use original embedding's mean vector or using interpolation when adding new tokens, enforcing consistency with these initial embeddings may also be semantically meaningless. This part would benefit from theoretical or empirical justification.

* W3: The paper introduces multiple components, but there is no quantitative report of training or inference time, nor a comparison with baselines such as TrajMoE or QT-Mob in computational efficiency.

* W4: The generation task uses the first two days of trajectories to predict the third day’s movements. However, weekday–weekend transitions (e.g., Friday to Saturday vs. Sunday to Monday) can involve very different behavioral dynamics, raising questions about the realism and generalizability of this setting.

* W5: The paper does not specify whether the four real-world city datasets are publicly available. The lack of dataset URLs or accessibility information limits reproducibility and independent verification of the results. If the datasets are proprietary, the authors should at least include one publicly available mobility dataset to enhance transparency and comparability.

* W6: The experiments primarily rely on Qwen as the LLM backbone. While these are reasonable choices, the lack of comparison with other widely adopted models (e.g., LLaMA, Gemma, or DeepSeek) limits the generality of the conclusions.

* W7: Merely Hit@1 is not enough. Mobility LLM uses Hit@1/5/20 and MRR. QT-Mob uses Hit@1/5/10 and NDCG@5/10.  GetNext uses Hit@1/5/10/20 and MRR. TrajMoE uses Hit@1/3/5

**Questions:**

* Q1: What's the definition of “geographically co-occurring locations.”? Does this term refer to trajectory co-occurrence or spatial proximity?

* Q2: How were the POI categories selected, and how to confirm that the taxonomy is consistent across cities?

* Q3: Have the authors experimented with alternative prompt templates or instruction formats for mapping between location IDs and descriptions in loc2id/id2loc tasks?

* Q4: Did the authors employ a prefix_allowed_tokens_fn or similar mechanism (as in QT-Mob) to restrict the LLM’s output to valid location tokens? If yes, was this constraint applied during both training and inference, or only in inference?

* Q5: Are all four cities using the same codebook and grid granularity? If so, how does the model handle locations that exist in one city but not in another? What ensures successful transfer when location IDs are not fully overlapping? (For example, using Atlanta with less than 2000 locations to predict/generate Chicago with more than 4000 locations)

* Q6: What does each learned codebook (location ID) look like in practice?

* Q7: Can the model guarantee that after adding new tokens, the generated trajectory will output eos all the time? Or you have to truncate the trajectory according to max token length or other thresholds?

I will reconsider my rating if all answered properly.

---

### Official Review · Reviewer_hZrC · 2025-10-25

**Soundness:** 3
**Presentation:** 2
**Contribution:** 2
**Rating:** 4
**Confidence:** 3

**Summary:**

This paper proposes a framework that adopts the semantic reasoning capabilities of Large Language Models (LLMs) to achieve a deeper understanding of human mobility. MoveFM-R features with three key components: a semantically enhanced location encoding mechanism that discretizes continuous geographic spaces using residual quantized variational autoencoders and aligns them with the LLM’s vocabulary; a progressive curriculum that guides the LLM from low-level trajectory description to high-level summarization of spatio-temporal patterns; and an interactive self-reflective reasoning process that iteratively refines generated trajectories under scenario-specific constraints through reinforcement learning. Experiments on real-world mobility datasets from four U.S. cities demonstrate that MoveFM-R outperforms both MFM-based and LLM-based baselines in prediction and generation tasks.

**Strengths:**

S1. The paper makes an important conceptual advance by proposing a unified paradigm that integrates MFMs with LLMs together. This synthesis offers a fresh research direction for mobility modelling.

S2. It is a trending attempt to leverage LLMs for mobility data analysis. It is interesting to develop methods that can have a better understanding of mobility patterns for downstream analytical tasks.

S3. Empirically, MoveFM-R demonstrates consistent improvements across a range of benchmark datasets and evaluation settings for both prediction and generation tasks.

**Weaknesses:**

W1. The proposed MoveFM-R framework integrates multiple indenpendent components, an MFM backbone, an RQ-VAE semantic encoder, a two-stage alignment module, a progressive curriculum for reasoning, and a GRPO-based reinforcement learning phase. This pipeline is computationally heavy and difficult to reproduce, requiring extensive GPU resources (A800 and A100 clusters) and multi-stage fine-tuning. This engineering overhead limits accessibility and scalability of the proposed framework.

W2. Although the paper emphasizes the “synthesis” between MFMs and LLMs, certain aspects of the design remain under-specified and somewhat unclear. It appears that the codebook is first derived and trained during the initial stage, and then subsequently passed to the MFM to generate latent representations in the later stage. This transition between stages feels unintuitive (i.e., the codebook is not utilized as inputs in subsequent stages), as the rationale behind the sequential dependency is not well explained.

W3.  Some parts of the paper’s technical details are questionable and could benefit from further clarification, including the codebook generation part and the initial token embedding part.

W4. This paper utilizes datasets from four cities. While this scale is nontrivial, it may not fully justify the characterization of the model as a “foundation” model, which typically implies training on orders of magnitude larger data. In addition, these datasets do not appear to be publicly available, which would be difficult and less beneficial for the research community to reproduce or extend the experiments.

**Questions:**

Q1. I don’t see clear differences between using RQ-VAE for one city and multiple cities. Since the codebook space is substantially larger than the number of distinct locations, it is not evident why multi-city training would yield a meaningful difference. Moreover, despite the claim that the codebook covers millions of locations, Appendix C indicates that each dataset contains only thousands of locations, which raises questions about the stated scale and its impact on performance.

Q2. In Section 3.1.2, the authors mention that the codebook tokens are not randomly initialized but derived from the mean of their constituent subword embeddings. However, the justification for this design choice is not clear. Given that the tokens appear as structured forms such as <a_x>/<b_x>/<c_x> (similar to special tokens in LLMs), it doesn’t make sense to me averaging unrelated subword embeddings contributes to meaningful initialization. Additionally, the term $y$, described as the “original pre-quantization semantic vector,” is not sufficiently explained.

Q3. Please response to other comments.

---

### Official Review · Reviewer_Cg52 · 2025-10-28

**Soundness:** 2
**Presentation:** 1
**Contribution:** 2
**Rating:** 4
**Confidence:** 5

**Summary:**

This paper proposes MoveFM-R, a Mobility Foundation Model built one Large Language Models to improve semantic reasoning and generalization in human mobility modeling. It introduces a semantic enhanced location encoding for bridging continuous and discrete spaces, a description-to-summarization curriculum for learning spatiotemporal reasoning, and a self-reflective reasoning loop for iterative trajectory generation.

**Strengths:**

- The paper tackles an emerging problem which is combining spatiotemporal mobility models with LLM reasoning. This is of growing interest in the foundation model community across different domains.
- The framework integrates multiple popular learning paradigms (SFT, alignment, GRPO).
- Experiments across different cities and different tasks demonstrate the performance of this model.

**Weaknesses:**

- The core modules mainly combine known techniques: RQ-VAE quantization from QT-Mob) description-summarization similar to existing instruction-tuning pipelines, and GRPO-based reflection borrowed from DeepSeek reasoning tasks. The integration lacks a unifying theoretical foundation or new algorithmic insight.
 - The projection from MFM embeddings to LLM space is a simple MLP without shared optimization. It is unclear whether meaningful cross-modal alignment occurs or whether the LLM merely learns to echo MFM outputs. There is no analysis of feature similarity or embedding alignment quality.
- The presentation contains excessive verbosity and marketing tone. Many passages use sweeping or promotional phrasing (e.g., “unlocking the full potential,” “pioneering a new paradigm”) that weakens scientific precision and makes the contribution appear overstated relative to its evidence.

**Questions:**

- How to verify that the LLM actually understands geographic semantics instead of memorizing coordinate–token pairs? More results regarding cosine similarity between semantically similar locations in embedding space would be helpful.
- Can the authors quantify how many edit iterations are typically needed, and show examples where reflection corrected implausible trajectories?
- How sensitive is the model to the codebook depth (number of quantization layers N) and vocabulary size? Does a larger or finer codebook improve or harm generalization?
- Any trajectory maps or language-conditioned examples to demonstrate the claimed interpretability?

---

### Meta-Review · Area_Chair_14dx · 2025-12-26

**Summary:**

Reviewers identified four primary areas for improvement: the incremental nature of the semantic components, the high resource requirements for reproducibility, the insufficiency of the evaluation metrics, and general presentation inconsistencies.  The authors did not submit a rebuttal and therefore did not engage with any of these questions. The absence of a rebuttal means that the authors did not engage with the reviewers’ questions. As a result, none of the concerns were addressed or clarified, and key issues regarding the soundness and contribution of the paper remain unresolved.

**Reviewer Concerns:**

The authors did not submit a rebuttal and therefore did not engage with any of these questions. The absence of a rebuttal means that the authors did not engage with the reviewers’ questions. As a result, none of the concerns were addressed or clarified, and key issues regarding the soundness and contribution of the paper remain unresolved.

**Reviewer Scores:**

The reviewers’ scores are expected to remain opposing the acceptance.

---

### Decision · Program_Chairs · 2026-01-26

Reject